# Enzyme Replacement Therapy in Mucopolysaccharidosis Type VII: A Three-Year Clinical Outcome Study of the First Taiwanese Case

**DOI:** 10.3390/diagnostics15040464

**Published:** 2025-02-14

**Authors:** Chung-Lin Lee, Chih-Kuang Chuang, Huei-Ching Chiu, Ya-Hui Chang, Yuan-Rong Tu, Yun-Ting Lo, Hsiang-Yu Lin, Shuan-Pei Lin

**Affiliations:** 1Department of Pediatrics, MacKay Memorial Hospital, No. 92, Sec. 2, Chung-Shan North Road, Taipei 10449, Taiwan; clampcage@gmail.com (C.-L.L.); g880a01@mmh.org.tw (H.-C.C.); wish1001026@gmail.com (Y.-H.C.); 2Institute of Clinical Medicine, National Yang-Ming Chiao-Tung University, Taipei 112304, Taiwan; 3International Rare Disease Center, MacKay Memorial Hospital, Taipei 10449, Taiwan; andy11tw.e347@mmh.org.tw; 4Department of Medicine, Mackay Medical College, New Taipei City 25245, Taiwan; 5Mackay Junior College of Medicine, Nursing and Management, Taipei 112021, Taiwan; 6Division of Genetics and Metabolism, Department of Medical Research, MacKay Memorial Hospital, Taipei 10449, Taiwan; mmhcck@gmail.com (C.-K.C.); likemaruko@hotmail.com (Y.-R.T.); 7College of Medicine, Fu-Jen Catholic University, New Taipei City 24205, Taiwan; 8Department of Medical Research, China Medical University Hospital, China Medical University, Taichung 40402, Taiwan; 9Department of Infant and Child Care, National Taipei University of Nursing and Health Sciences, Taipei 11219, Taiwan

**Keywords:** mucopolysaccharidosis VII, enzyme replacement therapy, β-glucuronidase deficiency, lysosomal storage disorder, hydrops fetalis

## Abstract

**Background and Clinical Significance:** Mucopolysaccharidosis type VII (MPS VII), an ultrarare lysosomal storage disorder caused by β-glucuronidase deficiency, presents significant therapeutic challenges. Given its extreme rarity and limited treatment experience in Asian populations, documenting long-term treatment outcomes is crucial for advancing clinical knowledge and improving patient care. **Case Presentation:** We report a 3-year follow-up of enzyme replacement therapy (ERT) in the first Taiwanese case of MPS VII. The patient, who initially presented with hydrops fetalis and developmental delay, was diagnosed at age 4 through genetic analysis, which revealed compound heterozygous variants of c.104C > A (p.Ser35Ter) and c.1454C > T (p.Ser485Phe) on the *GUSB* gene. ERT with vestronidase alfa was initiated at age 6. During the follow-up period, significant clinical improvements were observed, including elimination of oxygen dependency, with BiPAP needed only during sleep; changes in mobility, with 6-min walk test distance showing an initial decline from 130 to 70 m followed by partial recovery to 95 m after multiple orthopedic surgeries; and steady progression of growth parameters showed, with height increasing from 110 to 118 cm. Urinary glycosaminoglycan (GAG) levels measured by dimethylmethylene blue spectrophotometry decreased and stabilized. The patient’s cardiac and hepatic conditions remained stable, although splenomegaly persisted. No severe adverse events were reported during ERT. **Conclusions:** This case demonstrates the effectiveness and safety of long-term ERT in MPS VII, particularly in improving respiratory function and physical performance. Our experience highlights the importance of early diagnosis and treatment initiation, while providing valuable insights into the management of this ultrarare disease in the Asian population.

## 1. Introduction

### 1.1. Brief Overview of MPS VII and Its Rarity

Mucopolysaccharidosis type VII (MPS VII), also known as Sly syndrome, is an ultrarare autosomal recessive lysosomal storage disorder caused by pathogenic variants in the *GUSB* gene, which causes a deficiency in the β-glucuronidase enzyme [1,2]. This enzyme deficiency promotes the progressive accumulation of glycosaminoglycans (GAGs), including chondroitin sulfate, dermatan sulfate, and heparan sulfate, in multiple tissues and organs [3]. The prevalence of MPS VII varies from 0.02 to 0.24 per 100,000 live births worldwide [4]. Moreover, its clinical manifestations can range from severe forms presenting with non-immune hydrops fetalis to attenuated phenotypes, with common features including coarse facial features, skeletal dysplasia, hepatosplenomegaly, cognitive impairment, and cardiopulmonary complications [5]. Around half of the patients with severe MPS VII die before 1 year of age, highlighting the critical importance of early diagnosis and intervention [2]. Early diagnosis remains challenging but is crucial for optimal outcomes. When non-immune hydrops fetalis is detected prenatally, systematic screening including enzyme analysis and molecular testing should be considered, particularly given that approximately 40% of MPS VII cases present with hydrops fetalis [6]. Implementation of newborn screening programs could further facilitate early detection and intervention.

### 1.2. Current Treatment Options with a Focus on Enzyme Replacement Therapy (ERT)

Current treatment options for MPS VII primarily include ERT and hematopoietic stem cell transplantation (HSCT). ERT with vestronidase alfa (Mepsevii™, Ultragenyx Pharmaceutical Inc., Novato, CA, USA), which was approved by the Food and Drug Administration in November 2017, represents a significant advancement in the treatment of MPS VII [5]. Clinical studies have demonstrated ERT’s effectiveness in reducing urinary GAG levels and improving organomegaly [7]. However, ERT has certain limitations, one of which is the inability of the enzyme to effectively cross the blood–brain barrier, making it less effective for neurological manifestations [8]. In contrast, HSCT offers the potential advantage of enzyme delivery across the blood–brain barrier through donor-derived microglial cells but carries significant procedural risks [9,10]. Early initiation of treatment appears crucial for optimal outcomes, as demonstrated in clinical studies showing better response when therapy begins before extensive tissue damage occurs [11].

### 1.3. Importance of Long-Term Follow-Up Data

Long-term follow-up data for patients with MPS VII are crucial for several reasons. Given the exceeding rarity of this disease, with prevalence rates ranging from 0.02 to 0.24 per 100,000 live births [4], each patient’s longitudinal data contribute significantly to our understanding of disease progression and treatment effectiveness. Extended follow-up studies have revealed varying clinical trajectories and treatment responses, which have helped establish optimal therapeutic strategies [3]. Furthermore, long-term monitoring is essential for detecting potential late-onset complications and evaluating the sustained efficacy of treatments [2]. Such data are particularly valuable for understanding developmental outcomes and quality of life over time, especially in patients who begin treatment early in life [12]. Cumulative evidence from long-term follow-up studies has also proven instrumental in improving clinical guidelines and supporting the establishment of healthcare policies regarding the management of this ultrarare disease [2].

### 1.4. Significance of the First Case in Taiwan

The documentation of the first MPS VII case in Taiwan represents a significant milestone in research and management of this rare disease in Asia. As the first documented case among Taiwan’s population of approximately 23 million [13], it provides valuable insights into the genetic and clinical manifestations of MPS VII in the Asian population. This case contributes to the global understanding of the ethnic distribution of MPS VII and its potential genetic variations. Furthermore, as the first case of ERT managed in Taiwan, it establishes an important reference point for regional healthcare systems in treating ultrarare diseases and helps inform healthcare policies regarding orphan drugs in Asian healthcare contexts. The detailed documentation of this case also facilitates better awareness, recognition and an associated potential treatment option of MPS VII among healthcare providers in Taiwan and neighboring regions.

## 2. Methods

### Study Design and Patient Follow-Up

This single-case observational study documented a 3-year follow-up after ERT initiation. The standardized assessments included:

Regular Monitoring:-Quarterly: Growth parameters, physical examination, urinary GAG and enzyme activity measurements-Biannual: Cardiac (echocardiography) and hepatic evaluation (ultrasound)-Annual: Comprehensive skeletal and developmental assessment-Clinical Evaluations:-Physical performance: Six-minute walk test (6MWT), distance–saturation product (DSP)-Respiratory function: Continuous SpO2 monitoring, polysomnography-Growth monitoring: Height, weight, BMI using WHO growth charts-Biochemical markers: GAG levels, enzyme activity, liver function tests

The patient received vestronidase alfa (4 mg/kg) every two weeks with >95% compliance throughout the study period.

## 3. Case Presentation

### 3.1. Patient Information

#### 3.1.1. Demographics and Initial Presentation [13]

The patient, a female born in October 2015 at 28 weeks of gestation, presented with significant prenatal complications, including hydrops fetalis requiring four intrauterine ascites aspirations. She was born via cesarean section due to fetal distress, with a birth weight of 1934 g despite having generalized edema. She was a twin, with the other fetus having died at 10 weeks of gestation. The initial postnatal period was complicated by respiratory difficulties requiring ventilatory support and chronic lung disease associated with prematurity. Early developmental milestones were delayed, with notable facial dysmorphism becoming progressively apparent during the first year of life. Additional early manifestations included hearing impairment, laryngomalacia, and hepatosplenomegaly, representing the classic multisystemic involvement of MPS VII.

#### 3.1.2. Key Clinical Manifestations [13]

The patient exhibited multiple characteristic features of MPS VII. By age 4, she had developed distinctive facial dysmorphism with midface hypoplasia, a flat nasal bridge, and hypertelorism. Musculoskeletal manifestations included skeletal dysplasia with paddle-shaped ribs, dysplastic vertebral bodies, and bilateral hip dysplasia. Respiratory complications were significant, requiring nighttime BiPAP support during sleep and oxygen supplementation during daytime activities. Cardiopulmonary evaluation revealed atrial septal defect (ASD) II and mild valvular abnormalities. Growth parameters consistently tracked below normal, with height and weight falling below the 3rd percentile. Neurologically, she exhibited developmental delays but maintained the ability to walk independently. Ophthalmological examination revealed progressive corneal clouding, with hearing impairment persisting despite interventions.

#### 3.1.3. Diagnostic Journey (Genetic Testing Results) [13]

The diagnostic journey of our patient began at age 4 when she was referred for the evaluation of progressive dysmorphic features and developmental delays. Initial screening revealed elevated urinary GAGs (33.68 mg/mmol creatinine; normal range for age 2–17 years: <20.98 mg/mmol creatinine). Enzyme analysis showed normal activities for MPS I, II, IIIB, IVA, and VI. Whole exome sequencing using Illumina NovaSeq 6000 System (Control Software v1.7 and Real-Time Analysis Software v3.4.4, Illumina Inc., San Diego, CA, USA) identified the following compound heterozygous variants in the *GUSB* gene: c.104C > A (p.Ser35Ter), a known pathogenic variant, and c.1454C > T (p.Ser485Phe), a novel variant inherited from the mother. Segregation analysis confirmed the maternal inheritance of c.1454C > T and paternal inheritance of c.104C > A through grandmother testing. The diagnosis was confirmed by demonstrating severely reduced β-glucuronidase activity (0.00 μmol/g protein/h; normal range: 255.31–681.29 μmol/g protein/h).

### 3.2. Clinical Course and Treatment

#### 3.2.1. Pre-ERT Status (2015–2021) [13]

Prior to ERT initiation, the patient experienced progressive multisystemic manifestations. Her respiratory status was significantly compromised, requiring continuous oxygen supplementation (1–1.5 L/min) and BiPAP support for approximately 10 h daily. Growth parameters remained poor, with her height tracking consistently below the 3rd percentile (95 cm at age 6). Musculoskeletal involvement progressed with the development of thoracolumbar scoliosis and bilateral hip dysplasia requiring surgical intervention. Cardiac evaluation revealed ASD II with valve thickening, whereas hepatic assessment showed progressive hepatosplenomegaly with coarse liver texture on ultrasound. Regular monitoring of urinary GAG levels showed persistent elevation (DMB ratio: 33.68 mg/mmol creatinine; normal range for age 2–17 years: <20.98 mg/mmol creatinine), and enzyme activity remained undetectable. Despite these challenges, she maintained limited ambulatory ability with support but had significantly reduced exercise tolerance.

#### 3.2.2. ERT Initiation and Regimen

ERT was initiated in June 2021 after confirmation of diagnosis and coverage approval. Treatment with vestronidase alfa (Mepsevii™, Ultragenyx Pharmaceutical Inc., Novato, CA, USA) was administered at the standard dose of 4 mg/kg [11]. While this dosing regimen has led to significant clinical improvements, particularly in respiratory function and mobility, the persistent elevation of urinary GAG levels suggests potential room for therapeutic optimization. However, any dose modifications would require careful consideration within Taiwan’s healthcare regulatory framework for rare diseases. The initial infusions were conducted under close monitoring in the hospital setting, with infusion rates being gradually increased starting at 2.5% of the total volume for the first hour, increasing to 5% for the second hour, and then up to 10% for the remaining duration, if well tolerated, following the recommended protocol as described in Section 1.2. The patient’s clinical features and laboratory findings before and after ERT initiation are summarized in Table 1.

#### 3.2.3. Three-Year Follow-Up Results (2021–2024)

##### Respiratory Function Improvement (Table 1)

Daytime oxygen dependency was eliminated after the first year of treatment. BiPAP support was required only during sleep, with average usage decreasing from 10 to 8 h daily. Respiratory infection frequency significantly decreased.

##### Growth Parameters (Figure 1)

Height increased from 110 to 118 cm (below 3rd percentile) (Z-score: −2.8 SD to −2.5 SD). Weight improved from 16 to 22 kg (3rd to 15th percentile) (Z-score: −2.6 SD to −2.1 SD). BMI improved from 13.2 to 15.8 (Z-score: −1.9 SD to −1.6 SD). Head circumference increased from 49.1 to 50.1 cm (25th percentile) (Z-score: −0.67 SD to −0.58 SD). Annual growth velocity increased from 2.71 cm/year pre-ERT to an average of 3.5 cm/year.
Figure 1Growth parameters and clinical outcomes during the 3-year follow-up of enzyme replacement therapy (ERT). The image shows the progression of height, weight, and 6-min walk test (6MWT) distance from ERT initiation (2021/06) to the latest follow-up (11/2024).
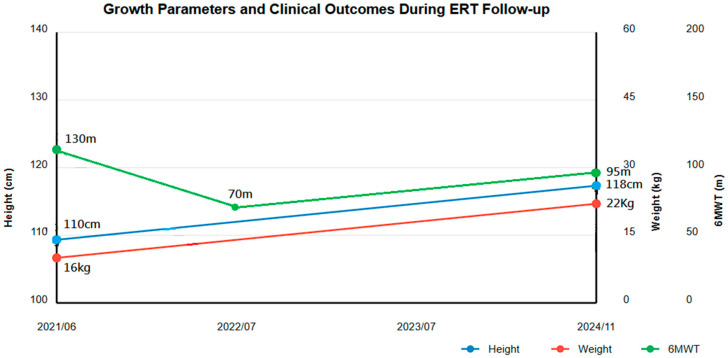


##### Physical Performance (Figure 1)

6MWT distance showed characteristic pattern:Baseline: 130 mInitial decline: 70 mLatest follow-up: 95 m

Despite not reaching baseline, improvement was noted considering multiple orthopedic surgeries during this period.

##### Other Organ Systems (Table 2)

Cardiac:Stable ASD II (0.35–0.40 cm)Mild interventricular septal thickeningProgressive aortic regurgitationEjection fraction maintained at 53–77%

Hepatic:Coarse liver textureLiver stiffness fluctuating between 0.68–1.40 m/sPersistent splenomegaly (9.92–10.93 cm)Tortuous splenic veins

**Table 2 diagnostics-15-00464-t002:** Serial cardiac evaluation findings during the 3-year follow-up of enzyme replacement therapy.

**Parameter**	June 2021	July 2022	November 2024
Cardiac Status *	ASD (0.40), EF 72.3%, Thick MV	ASD (0.339), EF 64%, Mild MR/AR	ASD (0.353), EF 77%, Mod MR/Mild AR

* ASD: atrial septal defect (size in cm); EF: ejection fraction; MV: mitral valve; MR: mitral regurgitation; AR: aortic regurgitation; Mod: moderate.

##### Quality of Life and Developmental Progress

Following ERT initiation, notable improvements in quality of life were observed. The elimination of daytime oxygen dependency significantly enhanced the patient’s ability to participate in school activities and social interactions. Sleep quality improved with optimized BiPAP support, as reported by caregivers. The patient showed progress in fine motor skills, particularly in activities like drawing and self-feeding. Communication skills also improved, though remaining below age-appropriate levels. While formal QoL assessments were not conducted, parent reports indicated increased independence in daily activities and improved social engagement.

### 3.3. Laboratory Findings (Figure 2)

#### 3.3.1. Urinary GAG Levels

Baseline 33.68 mg/mmol creatinine decreased to 15.21 mg/mmol creatinine, representing a 54.8% reduction.

#### 3.3.2. Enzyme Activity

From undetectable at baseline, enzyme activity levels (measured 7 days after ERT infusion) showed variable improvement:-3 months: 42.59 μmol/gm protein/h-Peak: 166.3 μmol/gm protein/h (18 months)-Latest: 57.42 μmol/gm protein/h

All enzyme activity measurements were consistently performed 7 days following ERT infusion to maintain standardization and allow meaningful comparison across timepoints.

#### 3.3.3. Other Biochemical Markers

Liver function: Mild transaminase elevation (aspartate aminotransferase, 41 IU/L and alanine aminotransferase, 31 IU/L) and gamma-glutamyl transferase (r-GT, 18 IU/L)Hematology: Mild leukopenia (WBC 3.9 × 10^3^/µL)Renal function: Normal creatinineLipids: Total cholesterol 109 mg/dL, triglycerides 62 mg/dLCardiac function: Ejection fraction 53–77%, progressive valvular disease with mild aortic and moderate mitral regurgitation

**Figure 2 diagnostics-15-00464-f002:**
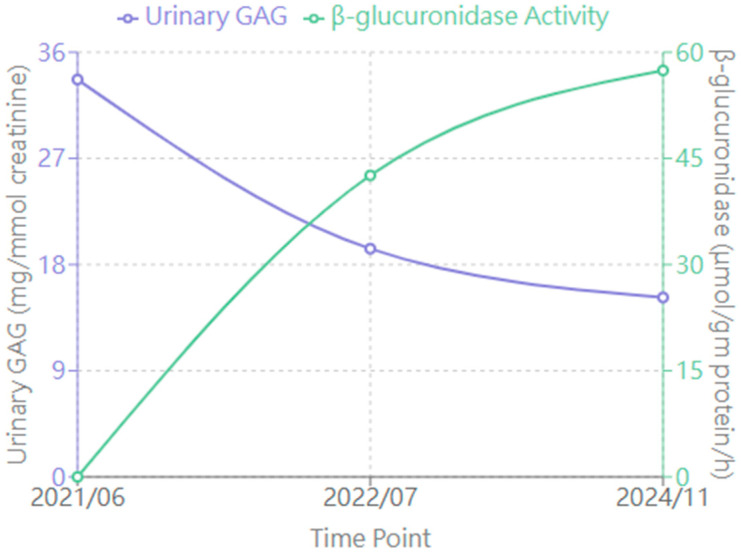
Changes in biochemical markers during enzyme replacement therapy (ERT). The graph shows urinary glycosaminoglycan (GAG) levels (mg/mmol creatinine, left *y*-axis, purple line) and β-glucuronidase activity (μmol/gm protein/h, right *y*-axis, green line) at three time points: before ERT (2021/06), after 1 year of ERT (2022/07), and at latest follow-up (2024/11). Normal ranges: urinary GAG < 20.98 mg/mmol creatinine; β-glucuronidase activity: 255.31–681.29 μmol/gm protein/h.

## 4. Discussion

The 3-year follow-up data demonstrate several clinically significant improvements following ERT initiation. The most remarkable improvement was in respiratory function, with the elimination of daytime oxygen dependency representing a major therapeutic milestone [14,15]. Growth parameters showed modest but steady improvement, with height increasing by 8 cm over 3 years and BMI improving from 13.2 to 15.8, marking a significant departure from the natural disease course [5]. The stabilization of cardiac and hepatic parameters, despite some progression in valve involvement, suggests that ERT may effectively slow disease progression in certain organ systems [2,3].

Our case demonstrates both similarities and notable differences when compared to previously reported MPS VII cases treated with ERT [4,16,17]. The initial presentation with hydrops fetalis aligns with approximately 40% of reported cases [6,7]. However, our patient’s survival beyond the critical first year represents a more favorable outcome than typically reported in the literature [8,9]. The response to ERT showed patterns comparable to those in phase III clinical trials of vestronidase alfa, particularly regarding urinary GAG reduction and respiratory improvement [10,11,12]. The documented improvements in mobility, while modest, appear more substantial than those reported in some adult-onset cases, possibly reflecting the advantages of earlier treatment initiation [18,19,20].

Several key challenges required careful clinical decisions during the 3-year follow-up period. Optimizing respiratory support posed an initial challenge, requiring careful BiPAP titration [21,22]. The management of progressive skeletal manifestations necessitated multiple surgical interventions, carefully timed between ERT infusions [23,24]. The standard dose of 4 mg/kg every 2 weeks was maintained with careful monitoring of biomarkers [25]. Although urinary GAG levels showed significant initial reduction, some fluctuation in enzyme activity levels necessitated continued close monitoring and highlighted the need for developing new biomarkers that might better reflect tissue-specific responses to therapy [13,14,15].

As the first documented case of MPS VII treated with ERT in Taiwan, our experience provides valuable insights into treatment response patterns among Asian populations [5,13]. The positive therapeutic response suggests comparable efficacy across ethnic groups, while our patient’s specific genetic variants highlight the importance of considering population-specific factors in treatment planning [26,27]. This case also demonstrates the critical role of healthcare policy in enabling access to treatments for ultrarare diseases [28,29]. The successful implementation of ERT was facilitated by Taiwan’s comprehensive rare disease policy framework, which could serve as a model for other Asian healthcare systems.

This case emphasizes that MPS VII should be considered in the differential diagnosis of non-immune hydrops fetalis, as early recognition could enable prompt initiation of ERT and potentially better outcomes [1,2,3]. Studies of MPS I, II, and VI have consistently demonstrated that initiation of ERT in infancy, particularly before significant tissue damage occurs, leads to better clinical outcomes [4,5]. Although long-term outcome data for early ERT in MPS VII remains limited due to its ultra-rare nature, emerging evidence suggests that early intervention may prevent or significantly delay the progression of various disease manifestations [6,7,8]. The experience from our case, where treatment initiation at age 6 still resulted in meaningful improvements, supports the therapeutic potential of ERT while highlighting the importance of early diagnosis [9,10,11,12].

## 5. Conclusions

This 3-year follow-up study of the first Taiwanese MPS VII patient treated with ERT demonstrated clinically meaningful improvements across multiple organ systems, with the most notable improvements being the progression from continuous oxygen dependency to requiring BiPAP support only during sleep, along with the modest but significant gains in growth parameters and physical performance. Our experience highlights three crucial aspects of MPS VII management: (1) the value of therapeutic intervention even after tissue damage has occurred, as evidenced by meaningful clinical benefits despite ERT initiation at age 6; (2) the necessity of a comprehensive, multimodal treatment approach given the variable responses across organ systems; and (3) the need for developing more refined monitoring tools based on observed biomarker patterns. This case emphasizes that MPS VII should be considered in the differential diagnosis of non-immune hydrops fetalis, as early recognition could enable prompt initiation of ERT and potentially better outcomes. Additionally, implementation of newborn screening programs could further facilitate early diagnosis and treatment initiation. Although our findings should be interpreted within the context of a single case report, this detailed longitudinal documentation of the first documented case of MPS VII treated with ERT in Taiwan provides valuable insights into treatment response patterns among Asian populations, demonstrates the feasibility of implementing ultrarare disease treatments in Asian healthcare systems under Taiwan’s comprehensive rare disease policy framework, and contributes significantly to the global understanding of long-term ERT outcomes.

## Figures and Tables

**Table 1 diagnostics-15-00464-t001:** Clinical features and laboratory findings in a Taiwanese girl with mucopolysaccharidosis type VII before and after enzyme replacement therapy (ERT).

Time Point	Growth Parameters [SDS]	Clinical Parameters	Laboratory Values †
Before ERT (June 2021)	Height: 95 [−3.2], Weight: 15 [−2.9], BMI: 16.6 [−1.8]	6MWT: 130 m, Continuous O2 1–1.5 L/min	β-glu: 0.00, uGAG:33.68
After ERT (July 2022)	Height: 110 [−2.8], Weight: 16 [−2.7], BMI: 13.2 [−2.1]	6MWT: 70 m, Night BiPAP only	β-glu: 42.59, uGAG: 19.34
Latest (November 2024)	Height: 118 [−2.6], Weight: 22 [−2.3], BMI: 15.8 [−1.6]	6MWT: 95 m, Night BiPAP only	β-glu: 57.42, uGAG: 15.21

6MWT: six-minute walk test; BiPAP: bilevel positive airway pressure; SDS: standard deviation score. † Normal ranges: urinary GAG (uGAG) < 20.98 mg/mmol creatinine; β-glucuronidase (β-glu): 255.31–681.29 μmol/gm protein/h.

## Data Availability

Data are contained within the article.

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
