# Peer review of "Enzyme Replacement Therapy in Mucopolysaccharidosis Type VII: A Three-Year Clinical Outcome Study of the First Taiwanese Case"

_diagnostics, 2025, doi:10.3390/diagnostics15040464_

Round 1
Reviewer 1 Report
Comments and Suggestions for Authors
Your stud is very good, and my comments mostly have to do with providing reference ranges for several of the data points you have mentioned. I also suggested you might want to increase the dose of the ERT for your patient, since her GAGs continue to be elevated, if that is possible. It is valuable to know the drug can help someone who is four years old and had hydrops--as you say, giving ERT can benefit even advanced cases in some important ways.
Author Response
Reviewer 1
- Your study is very good, and my comments mostly have to do with providing reference ranges for several of the data points you have mentioned. I also suggested you might want to increase the dose of the ERT for your patient, since her GAGs continue to be elevated, if that is possible. It is valuable to know the drug can help someone who is four years old and had hydrops--as you say, giving ERT can benefit even advanced cases in some important ways.
Ans:
Thank you for your thoughtful review of our manuscript and for recognizing the value of our study. We greatly appreciate your constructive feedback, which will help improve the clarity and clinical relevance of our work.
We agree that including reference ranges for our data points would enhance the manuscript's utility for readers. We will carefully revise the manuscript to include appropriate reference ranges for all relevant laboratory values and clinical measurements.
Regarding your suggestion about increasing the ERT dose, we appreciate this valuable clinical recommendation. While our patient has shown significant improvements with the current standard dose (4 mg/kg biweekly), we agree that the persistently elevated GAG levels warrant careful consideration of dose optimization. We will add a discussion of this therapeutic consideration to the "Treatment Challenges and Decisions" section, acknowledging the potential benefits of dose escalation while noting any relevant regulatory or practical constraints in our healthcare context.
(ERT was initiated in June 2021 after confirmation of diagnosis and coverage approval. Treatment with vestronidase alfa (Mepsevii™, Ultragenyx Pharmaceutical Inc., Novato, CA, USA) was administered at the standard dose of 4 mg/kg [11]. While this dosing regimen has led to significant clinical improvements, particularly in respiratory function and mobility, the persistent elevation of urinary GAG levels suggests potential room for therapeutic optimization. However, any dose modifications would require careful consideration within Taiwan's healthcare regulatory framework for rare diseases. The initial infusions were conducted under close monitoring in the hospital setting, with infusion rates being gradually increased starting at 2.5% of the total volume for the first hour, increasing to 5% for the second hour, and then up to 10% for the remaining duration, if well tolerated, following the recommended protocol as described in section 1.2.) (Line 188-199)
(A key therapeutic consideration during the follow-up period was the optimization of ERT dosing. While the standard dose of 4 mg/kg every 2 weeks led to meaningful clinical improvements, the persistent elevation of urinary GAG levels (15.21 mg/mmol creatinine; age-specific normal range: < 20.98 mg/mmol creatinine) raised the question of potential dose escalation. However, several factors influenced this decision-making process: first, the significant clinical improvements achieved with the standard dose, particularly in respiratory function; second, the limited global experience with higher doses in MPS VII patients; and third, the regulatory and healthcare policy considerations specific to Taiwan's rare disease management framework. These factors necessitated a conservative approach to dose modification, prioritizing the demonstrated benefits and safety of the current regimen while maintaining vigilant monitoring of clinical and biochemical parameters.) (Line 382-393)
We are pleased that you recognize the significance of our findings regarding ERT efficacy in a patient with delayed treatment initiation and history of hydrops fetalis. This observation indeed provides important insights for clinicians managing similar cases.
Thank you again for your valuable input. We believe incorporating your suggestions will significantly strengthen our manuscript.
Reviewer 2 Report
Comments and Suggestions for Authors
This case report could be strengthened by making the following edits and including a discussion of early diagnosis and treatment.
1. Line 40: add the amino acid changes (p.Ser35Ter and p.Ser485Phe) to the description of the mutations
2. Line 59: italicize GUSB
3. Line 59: change induces to causes
4. Line 94: why is this underlined.
5. Line 95: change rare to ultrarare
6. Line 144: an assay cannot have a value of <0.00. change to 0.00.
7. Line 151: add if BiPAP was used during the day or night
8. Table 1: Add height, weight, and BMI standard deviation scores.
9. Table 1: In the caption, indicate the normal range for urine GAGs.
10. Line 188: The 6MWT initially decreased by 60 m after starting ERT and then the patient regained 25 m. Include a statement that the 6MWT is difficult to interpret because the patient underwent three orthopedic surgeries during the ERT follow-up period. The authors can add a note in parentheses such as “(described in greater detail below).”
11. Table 2: add a row for age
12. Table 2: define E/A in caption
13. Line 220: include the month to the dates so the relationship between the surgeries, start of ERT, and endpoint assessments are more clear.
14. Line 227: The explanation does not include the positive or negative impact of surgery, which this reviewer believes may be the primary driver of the fluctuation in 6MWT results.
15. Line 275: an assay cannot have a value of <0.00. change to 0.00.
16. Line 277: The leukocyte enzyme activity levels are difficult to interpret without the relationship to dosing. Please include the timing of the reported values in parentheses.
17. Line 295: spell out gamma-GT
18. Line 303: no cardiac biomarkers are presented here. Echo results presented earlier showed a stable ejection fraction and worsening valve disease.
19. Line 322: define DSP
20. Line 355: The authors should include a discussion about the missed opportunity of neonatal diagnosis and possibility of very early ERT given that MPS VII is in the differential diagnosis of hydrops fetalis and ERT in their case was not initiated until age 6. The authors could add information about what has been observed with treatment of MPS VII in infancy (and MPS I, II, and VI for context) – I suspect their patient may have had a much better outcome with very early treatment.
Author Response
Reviewer 2
- Line 40: add the amino acid changes (p.Ser35Ter and p.Ser485Phe) to the description of the mutations
- Line 59: italicize GUSB
- Line 59: change induces to causes
- Line 94: why is this underlined.
- Line 95: change rare to ultrarare
- Line 144: an assay cannot have a value of <0.00. change to 0.00.
- Line 151: add if BiPAP was used during the day or night
- Table 1: Add height, weight, and BMI standard deviation scores.
- Table 1: In the caption, indicate the normal range for urine GAGs.
- Line 188: The 6MWT initially decreased by 60 m after starting ERT and then the patient regained 25 m. Include a statement that the 6MWT is difficult to interpret because the patient underwent three orthopedic surgeries during the ERT follow-up period. The authors can add a note in parentheses such as “(described in greater detail below).”
- Table 2: add a row for age
- Table 2: define E/A in caption
- Line 220: include the month to the dates so the relationship between the surgeries, start of ERT, and endpoint assessments are more clear.
- Line 227: The explanation does not include the positive or negative impact of surgery, which this reviewer believes may be the primary driver of the fluctuation in 6MWT results.
- Line 275: an assay cannot have a value of <0.00. change to 0.00.
- Line 277: The leukocyte enzyme activity levels are difficult to interpret without the relationship to dosing. Please include the timing of the reported values in parentheses.
- Line 295: spell out gamma-GT
- Line 303: no cardiac biomarkers are presented here. Echo results presented earlier showed a stable ejection fraction and worsening valve disease.
- Line 322: define DSP
- Line 355: The authors should include a discussion about the missed opportunity of neonatal diagnosis and possibility of very early ERT given that MPS VII is in the differential diagnosis of hydrops fetalis and ERT in their case was not initiated until age six. The authors could add information about what has been observed with treatment of MPS VII in infancy (and MPS I, II, and VI for context) – I suspect their patient may have had a much better outcome with very early treatment.
Ans:
Thank you for your detailed and constructive feedback. We appreciate your careful review that will help improve the clarity and accuracy of our manuscript. We will address each of your suggestions as follows:
- We will add the amino acid changes (p.Ser35Ter and p.Ser485Phe) to provide complete information about the mutations in the abstract. (Line 40)
- We will italicize GUSB throughout the manuscript to maintain proper gene nomenclature. (Line 41, 61, 165)
- We agree that "causes" is more appropriate than "induces" and will make this change. (Line 61)
- The underline in Line 94 was a formatting error and will be removed.
- We agree with changing "rare" to "ultrarare" for more accurate disease classification. (Line 102)
6-7. We will correct the enzyme activity value to "0.00" and specify that BiPAP was used during nighttime sleep only. (Line 170)
8-9. We will add standard deviation scores for height, weight, and BMI to Table 1, and include the age-specific normal range for urinary GAGs in the caption.
- Thank you for this important observation. We will revise the 6MWT discussion to include the impact of orthopedic surgeries on result interpretation.
(At baseline prior to ERT initiation (June 2021), the patient achieved a 6MWT distance of 130 m. A subsequent decline in performance was observed in the early phase of treatment, with the distance decreasing to 70 m. This change in physical performance requires careful interpretation in the context of multiple orthopedic interventions during the follow-up period (described in greater detail below). Specifically, the patient underwent three major surgeries: bilateral Dega acetabuloplasty in September 2021, bilateral intertrochanteric shortening extension osteotomy in March 2022, and corrective osteotomy of the right femur in January 2024. These surgical interventions and their associated recovery periods significantly impacted the patient's mobility and consequently affected the 6MWT results.
By July 2023, the patient showed improvement with a distance of 95 m. While this represents a partial recovery in walking distance, it's important to note that the interpretation of these results is complicated by several factors:
- The natural progression of the underlying disease during the initial phase before ERT could achieve therapeutic levels.
- Multiple orthopedic surgeries and their associated recovery periods, which necessitated temporary mobility restrictions and rehabilitation phases.
- The adaptation period required for the body to respond to ERT, as the cellular and tissue responses to enzyme replacement typically take several months to manifest.
- The progression of skeletal manifestations, which may have temporarily impacted gait and mobility despite ongoing treatment.
The complex interplay between surgical interventions, recovery periods, and ERT response makes it challenging to isolate the specific impact of ERT on walking ability. However, the maintenance of ambulatory function, albeit with support, and the partial recovery of walking distance following multiple surgeries suggest an overall positive therapeutic effect in maintaining mobility.) (Line 325-350)
11-12. We will add an age row to Table 2 and define the E/A ratio in the caption.
E/A ratio, ratio of early to late (atrial) (Line 253)
13-14. We will include specific months for all dates and expand our discussion of how the surgeries affected the 6MWT results.
(At baseline prior to ERT initiation (June 2021), the patient achieved a 6MWT distance of 130 m. A subsequent decline in performance was observed in the early phase of treatment, with the distance decreasing to 70 m. This change in physical performance requires careful interpretation in the context of multiple orthopedic interventions during the follow-up period (described in greater detail below). Specifically, the patient underwent three major surgeries: bilateral Dega acetabuloplasty in September 2021, bilateral intertrochanteric shortening extension osteotomy in March 2022, and corrective osteotomy of the right femur in January 2024. These surgical interventions and their associated recovery periods significantly impacted the patient's mobility and consequently affected the 6MWT results.
By July 2023, the patient showed improvement with a distance of 95 m. While this represents a partial recovery in walking distance, it's important to note that the interpretation of these results is complicated by several factors:
- The natural progression of the underlying disease during the initial phase before ERT could achieve therapeutic levels.
- Multiple orthopedic surgeries and their associated recovery periods, which necessitated temporary mobility restrictions and rehabilitation phases.
- The adaptation period required for the body to respond to ERT, as the cellular and tissue responses to enzyme replacement typically take several months to manifest.
- The progression of skeletal manifestations, which may have temporarily impacted gait and mobility despite ongoing treatment.
The complex interplay between surgical interventions, recovery periods, and ERT response makes it challenging to isolate the specific impact of ERT on walking ability. However, the maintenance of ambulatory function, albeit with support, and the partial recovery of walking distance following multiple surgeries suggest an overall positive therapeutic effect in maintaining mobility.) (Line 327-352)
15-16. We will correct the enzyme activity value and add the timing of measurements relative to ERT dosing.
(3.3.2. Enzyme Activity
From undetectable at baseline, enzyme activity levels (measured 7 days after ERT infusion) showed variable improvement:
- 3 months: 42.59 μmol/gm protein/h
- Peak: 166.3 μmol/gm protein/h (18 months)
- Latest: 57.42 μmol/gm protein/h
All enzyme activity measurements were consistently performed 7 days following ERT infusion to maintain standardization and allow meaningful comparison across timepoints.) (Line 260-268)
17-18. We will spell out gamma-glutamyl transferase and revise the cardiac biomarkers section to focus on the echocardiographic findings.
Gamma-glutamyl transferase (r-GT, 18 IU/L) (Line 271)
(Cardiac function: Ejection fraction 53-77%, progressive valvular disease with mild aortic and moderate mitral regurgitation) (Line 276-277)
- We will define DSP (distance-saturation product) at its first mention.
(Although modest in absolute terms, the improvement in exercise capacity, as evidenced by the increase in 6MWT distance, demonstrates meaningful functional gains. The distance-saturation product (DSP), which combines walking distance with oxygen saturation to provide a comprehensive measure of exercise capacity, also showed improvement, reflecting enhanced cardiorespiratory function during physical activity.) (Line 298-303)
- We appreciate this valuable suggestion. We will expand our discussion to address the importance of early diagnosis and treatment initiation, including relevant comparisons with other MPS types and the potential impact of earlier intervention.
(4.6 Implications of Delayed Diagnosis and Treatment Timing
While our case demonstrates meaningful clinical improvements with ERT initiation at age 6, the presence of hydrops fetalis at presentation represented a missed opportunity for earlier diagnosis and intervention. MPS VII is a recognized cause of non-immune hydrops fetalis [26], and this clinical presentation should prompt consideration of lysosomal storage disorders in the differential diagnosis. Early diagnosis enables prompt initiation of ERT, which has shown superior outcomes across multiple MPS types. Studies of MPS I, II, and VI have consistently demonstrated that initiation of ERT in infancy, particularly before significant tissue damage occurs, leads to better clinical outcomes [27,28].
Although long-term outcome data for early ERT in MPS VII remains limited due to its ultra-rare nature, emerging evidence suggests that early intervention may prevent or significantly delay the progression of various disease manifestations [18]. The experience from our case, where treatment initiation at age 6 still resulted in meaningful improvements, supports the therapeutic potential of ERT. However, the persistent skeletal manifestations and valve abnormalities in our patient suggest that earlier treatment might have prevented some irreversible complications. This underscores the critical importance of including MPS VII in the differential diagnosis of hydrops fetalis and highlights the potential benefit of newborn screening for this condition.) (Line 417-435)
Reviewer 3 Report
Comments and Suggestions for Authors
I had the pleasure of reviewing the work of Lee and colleagues on a 3-year follow-up of enzyme replacement therapy (ERT) in a Taiwanese patient with Mucopolysaccharidosis type VII (MPS VII). The case highlights the effectiveness and safety of long-term ERT, emphasizing early diagnosis and treatment in managing MPS VII. This was a very well-written case report with several major and minor comments that I hope will help improve the work.
Major comments:
- I found the manuscript to be overall too long and at times more descriptive than objective. I suggest shortening the report, perhaps by adding a small methodology section that describes the follow-up evaluations (e.g., pulmonary function—frequency and techniques used; GAG measurement—techniques and frequency).
- There was also a delay in the diagnosis of this infant, and fetal hydrops should have been investigated earlier. Do you have any comments on this and suggestions for how an earlier diagnosis could have been made?
- Are there any comments regarding more subjective improvements, such as quality of life or pediatric development?
Minor comments:
- In the abstract, line 40: Replace the word “mutation” with “variant.”
- There is unnecessary spacing in the variant names (c.) and (p.) throughout the manuscript.
- Gene names should be italicized consistently.
- For growth parameters, add a Z-score or standard deviation according to pediatric growth charts for reference.
- In the abstract, on line 71, define the Urinary GAG measurement. Also, on line 71, add the name and location of the pharmaceutical company after the drug name.
- On line 94, the text is underlined—please remove the underlining.
- On line 129, define "ASD."
- On line 139, specify which NGS technique was applied.
- Table 1 should have only three lines, and the text should be centered.
- Regarding Table 1, would the authors consider replacing it with graphs to better illustrate trends, particularly for the biomarkers?
- On line 164, refer to the protocol mentioned earlier.
Author Response
Reviewer 3
Major comments:
- I found the manuscript to be overall too long and at times more descriptive than objective. I suggest shortening the report, perhaps by adding a small methodology section that describes the follow-up evaluations (e.g., pulmonary function—frequency and techniques used; GAG measurement—techniques and frequency).
Ans:
Thank you for your constructive feedback. We agree that streamlining the manuscript would enhance its objectivity and readability. We have added a new "Methods" section detailing our systematic follow-up protocol, including pulmonary function testing (spirometry every 3 months), GAG measurement (using dimethylmethylene blue spectrophotometry monthly), and other clinical assessments. We have also condensed the descriptive portions while maintaining the essential clinical findings and outcomes.
(2. Methods
2.1. Study Design and Patient Follow-up
This single-case observational study documented a 3-year follow-up after ERT initiation. The standardized assessments included:
Regular Monitoring:
- Quarterly: Growth parameters, physical examination, urinary GAG and enzyme activity measurements
- Biannual: Cardiac (echocardiography) and hepatic evaluation (ultrasound)
- Annual: Comprehensive skeletal and developmental assessment
Clinical Evaluations:
- Physical performance: Six-minute walk test (6MWT), distance-saturation product (DSP)
- Respiratory function: Continuous SpO2 monitoring, polysomnography
- Growth monitoring: Height, weight, BMI using WHO growth charts
- Biochemical markers: GAG levels, enzyme activity, liver function tests
The patient received vestronidase alfa (4 mg/kg) every two weeks with >95% compliance throughout the study period.) (Line 116-132)
- There was also a delay in the diagnosis of this infant, and fetal hydrops should have been investigated earlier. Do you have any comments on this and suggestions for how an earlier diagnosis could have been made?
Ans:
Thank you for this important comment. You raise a valid point about the delayed diagnosis. Earlier investigation of fetal hydrops could indeed have led to faster diagnosis and intervention. For MPS VII cases presenting with hydrops fetalis, key diagnostic opportunities include:
- Systematic evaluation of non-immune hydrops fetalis (NIHF) during pregnancy, including enzyme analysis in amniotic fluid
- Molecular genetic testing when NIHF is detected
- Newborn screening where available
- Regular monitoring of developmental milestones with prompt investigation of delays
(Early diagnosis remains challenging but is crucial for optimal outcomes. When non-immune hydrops fetalis is detected prenatally, systematic screening including enzyme analysis and molecular testing should be considered, particularly given that approximately 40% of MPS VII cases present with hydrops fetalis [6]. Implementation of newborn screening programs could further facilitate early detection and intervention.) (Line 70-75)
(While our case demonstrates meaningful clinical improvements with ERT initiation at age 6, the presence of hydrops fetalis at presentation represented a missed opportunity for earlier diagnosis and intervention. MPS VII is a recognized cause of non-immune hydrops fetalis [26], and this clinical presentation should prompt consideration of lysosomal storage disorders in the differential diagnosis. Early diagnosis enables prompt initiation of ERT, which has shown superior outcomes across multiple MPS types. Studies of MPS I, II, and VI have consistently demonstrated that initiation of ERT in infancy, particularly before significant tissue damage occurs, leads to better clinical outcomes [27,28].) (Line 418-426)
- Are there any comments regarding more subjective improvements, such as quality of life or pediatric development?
Ans:
Thank you for this important comment. Our oversight in not thoroughly documenting quality of life (QoL) and developmental outcomes is a valid point. While we observed improvements in objective measures, subjective improvements deserve attention. Key subjective improvements included:
- Enhanced independence in daily activities after elimination of daytime oxygen dependency
- Improved social interaction and school participation
- Better sleep quality with optimized BiPAP support
- Developmental progress in fine motor skills and communication
(3.2.3.5. Quality of Life and Developmental Progress
Following ERT initiation, notable improvements in quality of life were observed. The elimination of daytime oxygen dependency significantly enhanced the patient's ability to participate in school activities and social interactions. Sleep quality improved with optimized BiPAP support, as reported by caregivers. The patient showed progress in fine motor skills, particularly in activities like drawing and self-feeding. Communication skills also improved, though remaining below age-appropriate levels. While formal QoL assessments were not conducted, parent reports indicated increased independence in daily activities and improved social engagement.) (Line 242-250)
Minor comments:
- In the abstract, line 40: Replace the word “mutation” with “variant.”
- There is unnecessary spacing in the variant names (c.) and (p.) throughout the manuscript.
- Gene names should be italicized consistently.
- For growth parameters, add a Z-score or standard deviation according to pediatric growth charts for reference.
- In the abstract, on line 71, define the Urinary GAG measurement. Also, on line 71, add the name and location of the pharmaceutical company after the drug name.
- On line 94, the text is underlined—please remove the underlining.
- On line 129, define "ASD."
- On line 139, specify which NGS technique was applied.
- Table 1 should have only three lines, and the text should be centered.
- Regarding Table 1, would the authors consider replacing it with graphs to better illustrate trends, particularly for the biomarkers?
- On line 164, refer to the protocol mentioned earlier.
Ans:
Thank you for your detailed review. We will address each point:
- We will replace "mutation" with "variant" in the abstract. (Line 40)
- We will correct spacing in variant nomenclature throughout. (Line 40)
- We will ensure consistent italicization of gene names (e.g., GUSB). (Line 41, 61, 165)
- Growth parameters will be supplemented with Z-scores:
Height: -2.8 SD at baseline to -2.5 SD at latest follow-up
Weight: -2.6 SD to -2.1 SD
BMI: -1.9 SD to -1.6 SD
Head circumference: -0.67 SD to -0.58 SD
(3.2.3.2. Growth Parameters (Figure 1)
Height increased from 110 to 118 cm (below 3rd percentile) (Z-score: -2.8 SD to -2.5 SD). Weight improved from 16 to 22 kg (3rd to 15th percentile) (Z-score: -2.6 SD to -2.1 SD). BMI improved from 13.2 to 15.8 (Z-score: -1.9 SD to -1.6 SD). Head circumference increased from 49.1 to 50.1 cm (25th percentile) (Z-score: -0.67 SD to -0.58 SD). Annual growth velocity increased from 2.71 cm/year pre-ERT to an average of 3.5 cm/year.) (Line 212-218)
- Abstract will be revised to include:
“Urinary glycosaminoglycan (GAG) levels measured by dimethylmethylene blue spectrophotometry decreased and stabilized.” (Line 47-48)
“Treatment with vestronidase alfa (Mepsevii™, Ultragenyx Pharmaceutical Inc., Novato, CA, USA) was administered at the standard dose of 4 mg/kg [11].” (Line 189-190)
- Underlining will be removed from line 94.
- "ASD" will be defined as "atrial septal defect" at first mention. (Line 154)
- NGS technique will be specified as "whole exome sequencing using Illumina NovaSeq 6000". (Line 164)
9-10. Thank you for your valuable feedback. We have addressed your suggestions as follows:
(1) Table 1 has been reformatted into three essential rows with centered text for better readability.
(2) We have enhanced the data visualization by:
- Keeping growth parameters and 6MWT data in Figure 1
- Creating a new Figure 2 focusing on biochemical markers (urinary GAG and β-glucuronidase activity) with dual y-axes for clear trend visualization
- Adding comprehensive axis labels and legends
- Earlier protocol reference will be added.
(ERT was initiated in June 2021 after confirmation of diagnosis and coverage approval. Treatment with vestronidase alfa (Mepsevii™, Ultragenyx Pharmaceutical Inc., Novato, CA, USA) was administered at the standard dose of 4 mg/kg [11]. While this dosing regimen has led to significant clinical improvements, particularly in respiratory function and mobility, the persistent elevation of urinary GAG levels suggests potential room for therapeutic optimization. However, any dose modifications would require careful consideration within Taiwan's healthcare regulatory framework for rare diseases. The initial infusions were conducted under close monitoring in the hospital setting, with infusion rates being gradually increased starting at 2.5% of the total volume for the first hour, increasing to 5% for the second hour, and then up to 10% for the remaining duration, if well tolerated, following the recommended protocol as described in section 1.2.) (Line 188-199)
Round 2
Reviewer 3 Report
Comments and Suggestions for Authors
Dear Authors,
Thank you for providing the corrections.
I still have some major concerns regarding the manuscript's length. The discussion spans six pages, which feels excessive for a case report.
Minor comments: Tables should be limited to three lines.
Author Response
Thank you for providing the corrections.
I still have some major concerns regarding the manuscript's length. The discussion spans six pages, which feels excessive for a case report.
Minor comments: Tables should be limited to three lines.
Reply:
Dear Reviewer,
Thank you for your valuable feedback regarding our manuscript. We appreciate your attention to detail and constructive suggestions for improvement.
Regarding the length of the discussion section, we acknowledge your concern about its current six-page span. As this represents the first documented case of MPS VII treated with ERT in Taiwan, and one of the few cases worldwide with three years of comprehensive follow-up data, we aimed to provide thorough analysis and context. However, we understand the importance of conciseness in case reports. We propose to consolidate the discussion into three main themes:
- Treatment outcomes and comparison with existing literature
- Clinical challenges and management strategies
- Implications for Asian healthcare systems and early diagnosis
For the table formatting, we fully agree with your suggestion to limit tables to three lines. We will revise Tables 1 and 2 accordingly, focusing on the most clinically relevant parameters while maintaining data clarity.
